# Automated ELISA On-Chip for the Detection of Anti-SARS-CoV-2 Antibodies

**DOI:** 10.3390/s21206785

**Published:** 2021-10-13

**Authors:** Everardo González-González, Ricardo Garcia-Ramirez, Gladys Guadalupe Díaz-Armas, Miguel Esparza, Carlos Aguilar-Avelar, Elda A. Flores-Contreras, Irám Pablo Rodríguez-Sánchez, Jesus Rolando Delgado-Balderas, Brenda Soto-García, Diana Aráiz-Hernández, Marisol Abarca-Blanco, José R. Yee-de León, Liza P. Velarde-Calvillo, Alejandro Abarca-Blanco, Juan F. Yee-de León

**Affiliations:** 1Delee Corp., Mountain View, CA 94041, USA; ever@delee.bio (E.G.-G.); ricardo@delee.bio (R.G.-R.); gladys@delee.bio (G.G.D.-A.); miguel.esparza@delee.bio (M.E.); caguilar@delee.bio (C.A.-A.); rolandodelgado@delee.bio (J.R.D.-B.); brenda@delee.bio (B.S.-G.); diana@delee.bio (D.A.-H.); marisol@delee.bio (M.A.-B.); jose.yee@delee.bio (J.R.Y.-d.L.); liza@delee.bio (L.P.V.-C.); 2Escuela de Ingeniería y Ciencias, Tecnologico de Monterrey, Monterrey 64849, NL, Mexico; 3Departamento de Bioquímica y Medicina Molecular, Facultad de Medicina, Universidad Autónoma de Nuevo León, San Nicolás de los Garza 64460, NL, Mexico; elda.florescn@uanl.edu.mx; 4Laboratorio de Fisiología Molecular y Estructural, Facultad de Ciencias Biológicas, Universidad Autónoma de Nuevo León, San Nicolás de los Garza 66455, NL, Mexico; iram.rodriguezsa@uanl.edu.mx

**Keywords:** SARS-CoV-2, COVID-19, spike protein, ELISA, antibodies, serology, on-chip, automated, microfluidics

## Abstract

The COVID-19 pandemic has been the most critical public health issue in modern history due to its highly infectious and deathly potential, and the limited access to massive, low-cost, and reliable testing has significantly worsened the crisis. The recovery and the vaccination of millions of people against COVID-19 have made serological tests highly relevant to identify the presence and levels of SARS-CoV-2 antibodies. Due to its advantages, microfluidic-based technologies represent an attractive alternative to the conventional testing methodologies used for these purposes. In this work, we described the development of an automated ELISA on-chip capable of detecting anti-SARS-CoV-2 antibodies in serum samples from COVID-19 patients and vaccinated individuals. The colorimetric reactions were analyzed with a microplate reader. No statistically significant differences were observed when comparing the results of our automated ELISA on-chip against the ones obtained from a traditional ELISA on a microplate. Moreover, we demonstrated that it is possible to carry out the analysis of the colorimetric reaction by performing basic image analysis of photos taken with a smartphone, which constitutes a useful alternative when lacking specialized equipment or a laboratory setting. Our automated ELISA on-chip has the potential to be used in a clinical setting and mitigates some of the burden caused by testing deficiencies.

## 1. Introduction

The coronavirus disease 2019 (COVID-19) is caused by the severe acute respiratory syndrome coronavirus 2 (SARS-CoV-2) and was officially declared as a pandemic by the World Health Organization (WHO) in March 2020. By July 2021, public records registered over 200 million infections, 4 million deaths, and 3 billion COVID-19 vaccine doses administered worldwide; additionally, several genetic variants have been identified so far [1,2,3]. Since its inception, the attempts to contain this disease have consisted of confinement and molecular diagnostics. Real-time reverse transcription-polymerase chain reaction (RT-qPCR) is considered the gold standard method to diagnose COVID-19, which is based on the amplification of SARS-CoV-2 genes (*N*, *E*, *RdRp*, *orf1a*, and *orf1b*) and its detection by fluorescent reporters from nasopharyngeal swab samples [4,5]. Other molecular methods, such as immunoassays directed to detect viral antigens or anti-SARS-CoV-2 antibodies, have proven to be of utmost importance for pandemic mitigation [6,7,8]. Understanding how the levels of anti-SARS-CoV-2 antibodies fluctuate in recovered COVID-19 patients and vaccinated individuals is fundamental to better understand the disease, especially when discrepancies have been found in previous reports [9,10,11,12]. Under the current circumstances, where millions of viral infections exist, it is undeniable that vaccination is of vital importance, and immunoassays have taken a particularly relevant role in identifying and monitoring patients’ immune responses over time. Therefore, several research groups have proposed various COVID-19-related immunoassays, which are mainly based on viral antigens, such as the spike protein [13], receptor-binding domain (RBD) [14], and nucleoprotein [15], and include different strategies ranging from the traditional enzyme-linked immunoassay (ELISA) to more complex microfluidic immunoassays [16,17,18].

Throughout the course of the COVID-19 pandemic, several diagnostic challenges have emerged, particularly regarding the millions of tests required to face the disease spreading. Microfluidic technologies are a promising approach that could solve some of those problems because they enable the integration and automation of complete diagnostic protocols in a single chip (i.e., lab-on-a-chip), including all the steps ranging from sample preparation to the detection and quantification of the analyte of interest [19]. A lab-on-chip provides the services of a traditional laboratory setting but with the advantages of miniaturization, reducing the volume of reagents and incubation times [20]. Some of the difficulties that have restricted the widespread use of these technologies in medical and biomedical applications are the manufacturing processes to mass-produce these microfluidic devices and their subsequent functionalization to attach biomolecules to their surface [21,22]. Despite this, various microfluidic devices have been developed in response to the problems posed by the COVID-19 pandemic. For example, in the diagnostic field, Fassy et al. reported a microfluidic qPCR capable of processing 192 samples in parallel and quantifying the expression of different SARS-CoV-2 viral genes [23], whereas for serological testing, Swank et al. developed a microfluidic immunoassay for the detection of anti-SARS-CoV-2 IgG antibodies with the capacity to analyze several samples using a minimal volume of reagents [24].

In this work, we developed an automated ELISA on-chip capable of detecting anti-SARS-CoV-2 antibodies in serum samples from COVID-19 patients and vaccinated individuals. In this first approach, we addressed some of the diagnostic challenges related to technical limitations in the existing tests, such as the requirement of expensive equipment and on-site detection. The fact that all the steps of the immunoassay are automated increases the reproducibility of the assay by preventing human error. Furthermore, the platform does not require specialized training for its operation, along with other advantages derived from the use of microfluidics, such as minimum sample manipulation, the use of a reduced volume of reagents, and the possibility of reading the results using the camera of a smartphone.

## 2. Materials and Methods

### 2.1. Sample Collection and Preparation

Blood samples were collected and provided according to the protocol approved by the Institutional Review Board of the Alfa Medical Center (AMCCI-TECCOVID-001). All the participants provided written informed consent before sample collection, which took place in a clinical laboratory that followed the guidelines established by Official Mexican Standards: NOM-007-SSA3-2011, NOM-087-SEMARNAT-SSA1-2002, NOM-010-SSA2-2010, NOM-006-SSA2-2013, and NMX-EC-15189 IMNC-2015. Blood samples were centrifuged at 1000× *g* for 10 min at 4 °C to separate the serum. In this work, a total of 22 serum samples were analyzed; 7 of them belonged to COVID-19 patients (samples from 3 and 7 months post-infection were available for 2 patients, while only samples from 7 months post-infection were available for the other 5 patients), 4 were from vaccinated volunteers (samples from 0 and 60 days post-vaccine were available for 2 patients, while only samples from 60 days post-vaccine were available for the other 2 patients), and 7 corresponded to healthy volunteers (2 samples were taken before the COVID-19 pandemic started). BSA and anti-spike-SARS-CoV-2 pAb (Sino Biological Inc., Chesterbrook, PA, USA) were included as a negative and positive control, respectively. The serum samples provided by the Alfa Medical Center were previously classified after performing qRT-PCR (Viasure SARS-CoV-2 S gene Real-Time PCR Detection Kit; CerTest Biotec SL., Zaragoza, Spain) and serological tests (Realy 2019-NCOV IgG/IgM Test; Hangzhou Realy Tech Co., Ltd., Hangzhou, China). Infected patients were selected after testing positive for COVID-19 using the qRT-PCR assay. In addition, the presence of anti-SARS-CoV-2 antibodies (IgG/IgM) was confirmed after performing serological tests 3 (when available) and 7 months following infection. Healthy and vaccinated volunteers were selected after testing negative for COVID-19 using the qRT-PCR assay and without detecting the presence of anti-SARS-CoV-2 antibodies (IgG/IgM). All procedures involving human participants were performed in accordance with the 1964 Declaration of Helsinki and its later amendments or comparable ethical standards.

### 2.2. Traditional ELISA on a Microplate

A traditional ELISA was performed using a 96-well microplate (Corning Inc., Tewksbury, MA, USA) to compare those results against the ones obtained with our automated ELISA on-chip. Firstly, 100 µL of a PBS suspension containing 1 µg/mL of the complete spike protein (Sino Biological Inc., Chesterbrook, PA, USA) was deposited in each well, followed by a 1 h incubation at room temperature. Afterward, three washes were made using a wash buffer (WB = PBS containing 0.05% Tween^TM^ 20 (Thermo Fisher Scientific, Waltham, MA, USA). Blocking was made by incubating 200 µL of 5% skim milk (Sigma-Aldrich, Burlington, MA, USA) at room temperature for 1 h. Subsequently, another round of three washes was carried out using the WB. Then, the serum samples (1:100 dilution) were added to the microplate and incubated for 1 h at room temperature to later be washed three times with WB. Next, a 1 h incubation of 100 µL of anti-human IgG conjugated with HRP (1:15,000 dilution; Thermo Fisher Scientific, Waltham, MA, USA) was performed, at room temperature, to identify the presence of anti-spike antibodies, followed by three washes with WB. Finally, 100 μL of 1-Step^TM^ Ultra TMB-ELISA (Pierce Biotechnology Inc., Rockford, IL, USA) was used to reveal the reaction, and the reaction was stopped by adding 100 μL of 1 M H_2_SO_4_.

### 2.3. Assay’s Methodology and Experimental Setup of the Automated ELISA On-Chip

The methodology followed in the automated ELISA on-chip assay and a diagram that illustrates the experimental setup implemented are displayed in Figure 1A,B, respectively. The reagents used and the conditions at which these were passed through the microfluidic device are specified in Table 1 and Appendix A. Commercially available microfluidic instrumentation was used to assemble the experimental setup that enabled the automation of our ELISA on-chip assay from antigen immobilization to the detection of anti-SARS-CoV-2 antibodies. A PS microfluidic device with four straight channels (50 µL volume capacity/channel; microfluidic ChipShop, Jena, Germany), a flow control unit (Zen Fluidics, Laredo, TX, USA), a 12/1 bidirectional microfluidic rotary valve (Zen Fluidics, Laredo, TX, USA), a microfluidic valve controller (Zen Fluidics, Laredo, TX, USA), a set of 4 pinch valves (Zen Fluidics, Laredo, TX, USA), and a set of four 3/2-way switching valves (Zen Fluidics, Laredo, TX, USA) were the main components that constitute this setup. All protocol steps were programmed using Zen Lab software (Zen Fluidics, Laredo, TX, USA), which also served to control all the components of the setup. All the pneumatic, fluidic, and electrical connections are also specified in the diagram depicted in Figure 1B and Appendix A. Additionally, a photograph of the experimental setup is shown in Figure 2.

### 2.4. Data Analysis

The colorimetric reactions obtained when performing the traditional ELISA and our automated ELISA on-chip were analyzed with two different methods. The first one used a microplate reader (BioTek Instruments, Winooski, VT, USA) and measured the reaction at 450 nm. The second approach was based on taking photos with a commercial smartphone (12 megapixel camera resolution) of the microplate and microfluidic devices, under the same lighting conditions, for further color intensity analysis using the color histogram plugin of the ImageJ software (Bethesda, MD, USA) to assign a value to each reaction.

### 2.5. Statistical Analysis

The Kolmogorov–Smirnov test was used to determine if our data followed a normal distribution. The results showed non-parametric behavior; therefore, the statistical analysis was performed with the Wilcoxon test, using a *p*-value < 0.01 and IBM SPSS Statistics 27 software (IBM Corporation, Armonk, NY, USA). All samples were run in triplicate. DataGraph version 4.7 (Visual Data Tools Inc., Chapel Hill, NC, USA) and Microsoft Excel (Microsoft Corporation, Redmond, WA, USA) were employed to graph the results.

## 3. Results and Discussion

### 3.1. Immunoassays’ Comparison

To validate our results, we performed a comparison between the traditional ELISA and our automated ELISA on-chip, in which anti-SARS-CoV-2 antibodies were detected. For each microfluidic device, four samples were processed in parallel in the same run. For both immunoassay formats, the colorimetric reactions derived from the horseradish peroxidase were analyzed with a microplate reader (Figure 3A). The 22 samples used in this work were collected from COVID-19 patients, vaccinated individuals, and healthy volunteers. The results showed a non-parametric distribution; therefore, the Wilcoxon test was employed for the statistical analysis, indicating that there was not a statistically significant difference between both immunoassays (*p*-value > 0.01). In addition, to corroborate these results, a Bland-Altman plot was carried out to compare the mean difference between the traditional ELISA and our automated ELISA on-chip. The plot indicated that there were no substantial differences between the measurements obtained for both formats when the colorimetric reactions were analyzed by a microplate reader at 450 nm using 99% limits of agreement (±2.575 SD).

### 3.2. Detection of Anti-SARS-CoV-2 Antibodies Post-Infection

We also monitored the presence of anti-SARS-CoV-2 antibodies in two patients diagnosed with COVID-19 at 3 and 7 months post-infection. In these patients, a decrease in absorbance was observed in samples collected 7 months after infection compared to the samples collected 3 months post-infection. Furthermore, for the other five patients, only samples from 7 months post-infection were available, which showed comparable or even higher antibody levels. The data obtained from both immunoassay formats, the traditional ELISA, and the automated ELISA on-chip, are displayed in Figure 3B.

The immune response after COVID-19 infection is heterogeneous among subjects, whereas, for diseases caused by other coronaviruses, such as SARS-CoV and MERS-CoV, antibodies can be detected up to 34 weeks after being infected [25,26]. As an example, Labriola et al. reported a decline in anti-spike and anti-nucleocapsid IgG levels in patients that tested positive for COVID-19 3 months after infection, while Dan et al. reported stable levels of anti-SARS-CoV-2 antibodies after 8 months from viral infection. Despite this heterogeneity, the immune response seen in patients after infection agrees with what has been described in other reports [27,28,29,30].

### 3.3. Detection of Anti-SARS-CoV-2 Antibodies Post-Vaccine

In this study, we also analyzed serum samples from four vaccinated individuals. Blood was collected and processed 60 days after vaccination. To assess the increase in anti-SARS-CoV-2 antibodies after inoculation, we collected two samples from two of those individuals before they were vaccinated. As seen in Figure 3C, using both assay formats, high levels of anti-SARS-CoV-2 antibodies were detected in the four samples extracted 60 days after vaccination. Furthermore, the immune response triggered by the administration of this biologic was observed in the two patients that had samples available before and after vaccination.

In an effort to control this pandemic, different immunization strategies have been developed. Currently, the FDA has approved three different vaccines based on either adenovirus or lipidic complexes containing mRNA molecules. In a recent work by Jalkanen et al., an increase in the production of anti-SARS-CoV-2 IgG antibodies was observed at six weeks after the administration of the mRNA vaccine (BNT162b2). Additionally, Stephenson et al. reported a detectable increase in anti-SARS-CoV-2 antibodies eight days after the administration of the adenoviral Ad26.COV2.S vaccine, detecting higher concentrations of these antibodies 57 and 71 days after inoculation. The results obtained in this work are in line with the results reported in other studies where these biologics were assessed, which showed an increase in anti-SARS-CoV-2 antibodies after the application of a vaccine [31,32,33].

### 3.4. Image Analysis of Colorimetric Reactions

Diagnostic tests usually require expensive equipment and/or specialized technicians, which complicates its implementation in underdeveloped communities. Therefore, we propose using photographs taken with a commercial smartphone to perform the analysis of the resulting colorimetric reaction as a simple and low-cost alternative to microplate readers. We were able to detect anti-SARS-CoV-2 antibodies in serum samples from COVID-19 patients and vaccinated individuals using both immunoassay formats and assign a value to each reaction by analyzing the color intensity of every picture taken. Figure 4A visually describes the analysis process, while pictures of the resulting reactions when performing the traditional ELISA and our automated ELISA on-chip are shown in Figure 4B. Since both results, the ones obtained from the microplate reader and the analysis with ImageJ, showed a non-parametric distribution, the Wilcoxon test was used for the statistical analysis, revealing that there were no statistically significant differences between these two approaches. These results are displayed in Figure 4C. To further corroborate these results, a Bland-Altman plot was carried out to compare the mean difference between the traditional ELISA and our automated ELISA on-chip. The plot indicated that there were no substantial differences between the values obtained with both immunoassay formats after ImageJ analysis when 99% limits of agreement (±2.575 SD) were used.

There is a clear trend toward the development of point-of-care (POC) diagnostic devices to mitigate healthcare deficiencies and improve the health conditions of people living in underserved communities. Its popularization is largely due to its compatibility to be mass produced, ease of operation, small footprint, lower costs, and the capacity to rapidly generate accurate and reliable results. To date, numerous POC devices have been developed for their use in a broad range of medical and biomedical applications, including the detection of infectious diseases, such as Zika [34], Ebola [35], and now, COVID-19 [36]. Furthermore, various devices have been cleared by FDA and are commercially available [37].

There are still some unknowns regarding the immune response to COVID-19 and its vaccines. For a better understanding, we need massive seroprevalence studies. Currently, we can find many traditional ELISA assays on the market, but at high costs that encompass prolonged protocols and require specialized staff to perform these tests; because of this, a reliable, sensitive, high-throughput, low-cost, and automated method for quantitatively measuring anti-SARS-CoV-2 antibodies is extremely necessary. In order to target this need, in this manuscript, we propose an automated ELISA on-chip to detect anti-SARS-CoV-2 antibodies in serum samples that aim to fulfill these requirements. For future iterations, the design of the microfluidic device that uses our platform can be improved to carry out an ELISA on-chip where more samples can be processed in parallel. Being able to process more samples simultaneously is an essential requirement in a clinical setting, hence the importance of assessing a design that could increase the number of samples that can be processed in parallel. In addition, we can optimize the amount of reagents needed and the incubation times to further reduce the duration of the immunoassay. Finally, this platform can be further validated using more samples from patients with COVID-19 or by detecting analytes relevant to other diseases.

## 4. Conclusions

Now more than ever, seroprevalence information has become highly relevant to monitor the behavior of COVID-19; however, in order to generate this data, a massive amount of immunoassays must be performed. The most critical factors that make it difficult to carry out diagnostic tests on a large scale are their related costs, sensitivity, specificity, and portability. A traditional ELISA presents several limitations, such as the long processing times due to the repetitive washing and incubation steps and the high costs of the reagents and equipment involved.

Microfluidic technologies can enable the integration and automation of complete di-agnostic protocols in a single chip, making them a truly advantageous option for mass testing, particularly in a pandemic setting. This manuscript presented a novel format for the detection of anti-SARS-CoV-2 antibodies by using an automated immunoassay on-chip while comparing it to the gold standard methodology utilized for immunoassays. Our work revealed that there were no significant differences between the results of both immunoassays, which were obtained from: (1) a microplate reader, the most common instrument to measure absorbance in immunoassays, and (2) an image processing software that analyzed a picture taken with a smartphone. In addition to being reliable, this last method constitutes a fast, easy, and low-cost alternative for analyzing colorimetric assays, especially in POC testing when a microplate reader or specialized technicians are not available in underserved communities.

## Figures and Tables

**Figure 1 sensors-21-06785-f001:**
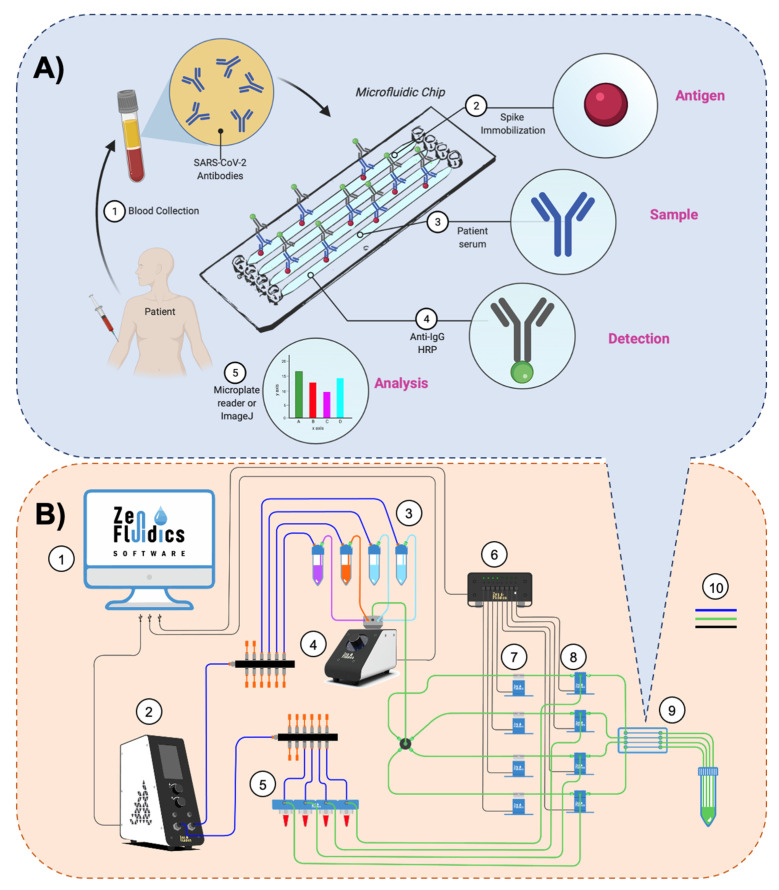
Assay’s methodology and experimental setup. (**A**) The methodology followed in our automated ELISA on-chip assay included the steps listed below: (**1**) blood collection and serum extraction; (**2**) on-chip immobilization of viral antigens (SARS-CoV-2 spike protein); (**3**) pumping serum samples through the microfluidic device (if present, anti-SARS-CoV-2 antibodies interact with the immobilized SARS-CoV-2 spike proteins); (**4**) pumping anti-IgG-HRP through the microfluidic device to detect IgG antibodies; and (**5**) analysis of the colorimetric reaction, either by recovering the resulting reaction and performing the analysis with a microplate reader or by carrying out a color intensity analysis using ImageJ software. (**B**) Diagram of the experimental setup assembled for our automated ELISA on-chip. The components are: (**1**) software that controls all the peripheral devices of the setup; (**2**) flow control unit; (**3**) ELISA reagents (wash buffer, blocking buffer, and anti-IgG-HRP conjugate antibody); (**4**) 12/1 bidirectional microfluidic rotary valve; (**5**) serum samples; (**6**) microfluidic valve controller; (**7**) pinch valves; (**8**) 3/2-way switching valves; (**9**) microfluidic device; and (**10**) connections between components and reservoirs (color code: blue is for pneumatic connections, green is for fluid connections, and black is for electrical connections).

**Figure 2 sensors-21-06785-f002:**
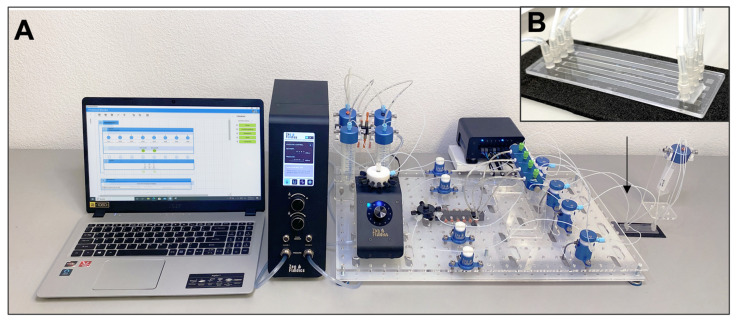
(**A**) Photograph of the experimental setup of our automated ELISA on-chip. (**B**) Picture of the microfluidic device used in our setup.

**Figure 3 sensors-21-06785-f003:**
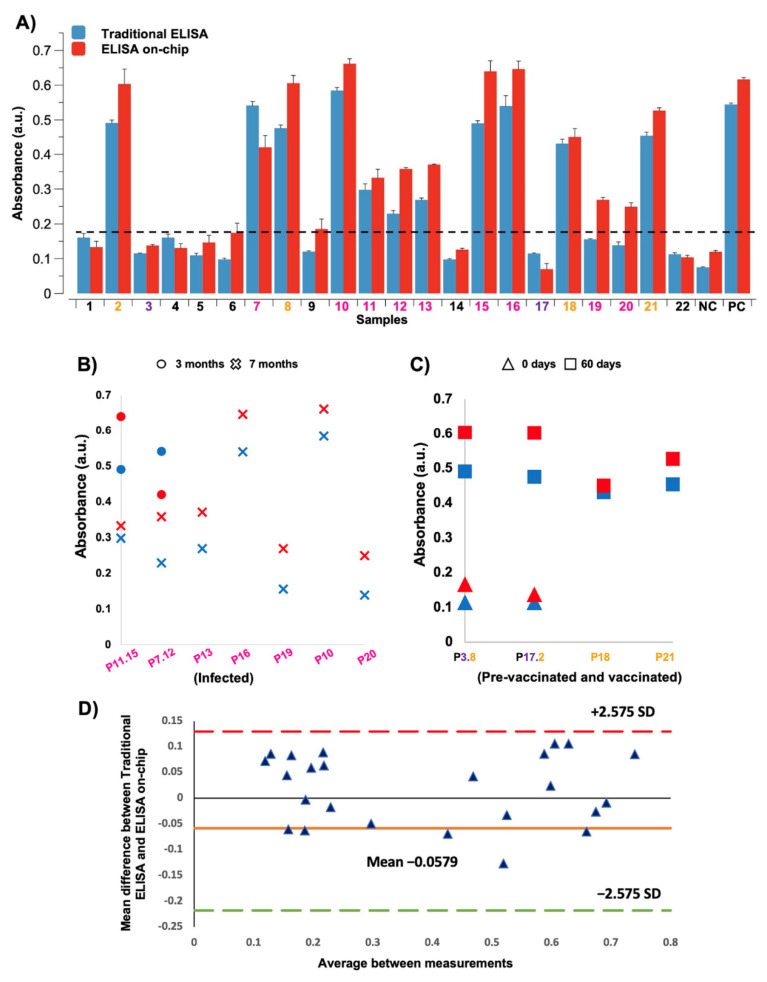
Absorbance comparison between the traditional ELISA and our automated ELISA on-chip. The resulting colorimetric reactions from both immunoassay formats were analyzed with a microplate reader. (**A**) Measured absorbance of the 22 serum samples, 1 negative control (NC), and 1 positive control (PC). The sample number is colored according to the following code: healthy subjects in black, pre-vaccinated subjects in purple, vaccinated subjects in orange, and infected subjects in pink. The dotted line indicates the absorbance limit measured in serum samples from healthy subjects. (**B**) Measured absorbance of the serum samples collected from COVID-19 patients 3 months (circle) and 7 months (cross) after infection. The sample color and patient code are in accordance with the one presented in Figure 3A. Patients with a code containing 2 numbers provided two different samples (e.g., patient P11.15 provided samples 11 and 15). (**C**) Measured absorbance of the serum samples collected from vaccinated individuals prior to being vaccinated (triangle) and 60 days after vaccination (square). The sample color and patient code are in accordance with the one presented in Figure 3A. Patients with a code containing two numbers provided two different samples (e.g., patient P3.8 provided samples 3 and 8). (**D**) Bland-Altman plot that shows the mean difference between the measurements obtained from the traditional ELISA and ELISA on-chip when the colorimetric reactions were analyzed by a microplate reader at 450 nm. For this analysis, 99% limits of agreement (±2.575 SD) were used.

**Figure 4 sensors-21-06785-f004:**
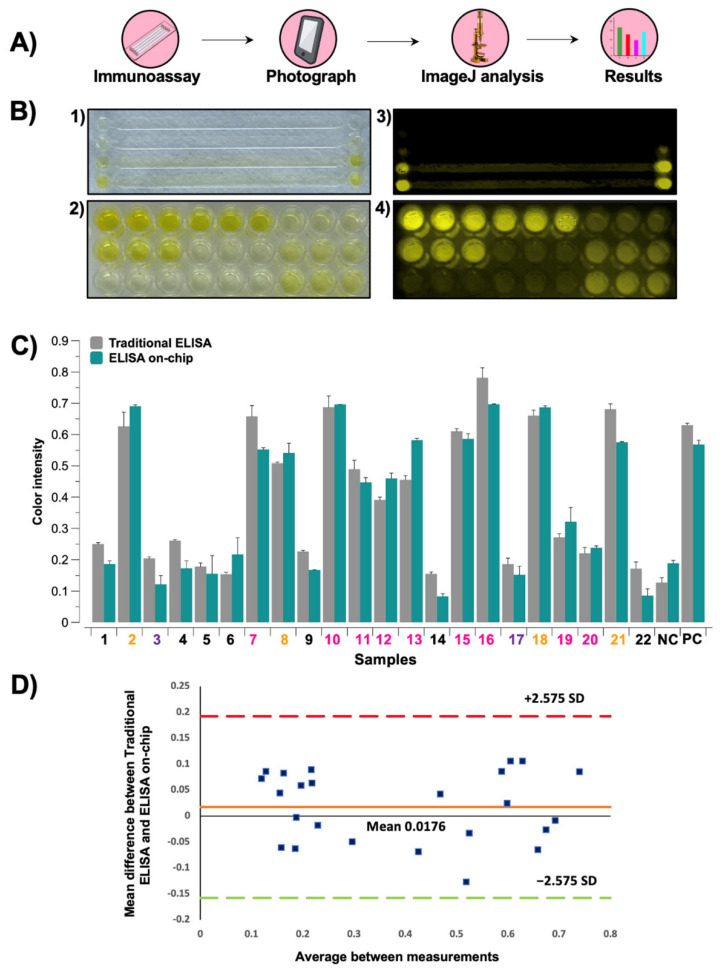
(**A**) Analysis methodology: First, the immunoassay is performed. Then, a photograph of the substrate, either a microplate or a microfluidic device, is acquired with a smartphone, followed by a color intensity analysis of the resulting colorimetric reactions using ImageJ software or any other similar software. Finally, the results are plotted and compared against the ones obtained from other samples. (**B**) Photographs of the resulting reactions when performing (**1**) our automated ELISA on-chip, (**2**) the traditional ELISA, and (**3**,**4**) its corresponding processed images. (**C**) Comparison of the results obtained by image analysis of the colorimetric reactions with both immunoassay formats from the 22 serum samples, 1 negative control (NC), and 1 positive control (PC). The sample number is colored according to the following code: healthy subjects in black, pre-vaccinated subjects in purple, vaccinated subjects in orange, and infected subjects in pink. (**D**) Bland-Altman plot of the results obtained from ImageJ analysis using both immunoassay formats that shows the mean difference between the measurements obtained from the traditional ELISA and ELISA on-chip. Here, 99% limits of agreement (±2.575 SD) were used.

**Table 1 sensors-21-06785-t001:** Established protocol for our automated ELISA on-chip.

Step	Reagent	On-Chip Flow Dynamics
1. Antigen immobilization	1 μg/mL spike protein in PBS	500 μL/min (30 s)–incubation (1 h/RT)
2. Wash	0.05% Tween-20^TM^ in PBS	500 μL/min (30 s)
3. Blocking	5% skim milk in PBS	500 μL/min (1 min)–incubation (30 min/RT)
4. Wash	0.05% Tween-20^TM^ in PBS	500 μL/min (1 min)–3 times
5. Pumping of samples	Serum (diluted in PBS 1:100)	500 μL/min (1 min)–incubation (50 min/RT)
6. Wash	0.05% Tween-20^TM^ in PBS	500 μL/min (1 min)–3 times
7. Anti-IgG-HRP binding	Anti-IgG-HRP (diluted in PBS 1:15,000)	500 μL/min (30 s)–incubation (50 min/RT)
8. Wash	0.05% Tween-20^TM^ in PBS	500 μL/min (1 min)–3 times
9. HRP reaction	TMB-ELISA	50 μL–incubation (3 min/RT)
10. Reaction halt	1 M H_2_SO_4_	50 μL (final reaction)

## Data Availability

Data is contained within the article and Appendix A. Additional raw data relevant to this study and/or related to clinical samples are available on request from the corresponding author. The clinical data are not publicly available due to ethical reasons related to the preservation of the privacy of volunteers.

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
