# Peer review of "Automated ELISA On-Chip for the Detection of Anti-SARS-CoV-2 Antibodies"

_sensors, 2021, doi:10.3390/s21206785_

Round 1
Reviewer 1 Report
In this communication, González-González et al., present an interesting project of ELISA lab-on-chip approach for the measurement of SARS-CoV-2 anti-spike IgG.
The topic of this study is relevant in the context of COVID-19 pandemic as it could provide alternative strategies for the diagnostic of COVID-19 and/or for sero-epidemiological studies. This approach could be useful to use lower reagents and sample volumes than traditional ELISA.
The authors did develop a tradional ELISA and lab-on-chip ELISA for measurement of COVID-19 anti-spike IgG with a microplate reader or a smartphone camera. They validated this approach with 22 human serums.
Strengths of this article is its relevance in the pandemic context, its feasibility due to the use of polystyrène and universal microfluidic systems.
However, it is difficult to review this article as the objective is unclear. In my view, the results only demonstrate that both ELISAs were able to to detect anti-spike IgG. It is barely possible to conclude on the clinical significance of these results as this study lacks a correct control group, lacks details regarding classification of patients has "infected", has a small sample size and lacks a reference immunoassay (commercial or published). Results do not allow to conclude on potential benefits of lab-on-chip ELISA's vs traditional ELISA.
The results are promising but the construction and content of the manuscript should be revised.
In my view, the authors should adress some major points:
-
Introduction : it is unclear what it the state of the art regarding SARS-CoV-2 diagnostic using microfluidic devices.
-
line 86 : The objective of this study is not clearly described. It is unclear if the objective was to develop a lab-on-chip ELISA, to demonstrate its agreement with ELISA results or to demonstrate its advantages over ELISA on plate. It is of limited interest to compare lab-on-chip results to another assay (ELISA) which has not been validated elsewhere.
-
line 122 : 5 healthy patients sampled during COVID-19 pandemic can have anti-spike Ab due to asymptomatic/paucy-symptomatic COVID-19. Therefore, they cannot be used to define average ELISA ratio for healthy patients.
-
page 4 line 124 : PCR cannot be used to confirm serological results because PCR stays positive for 14 days after onset of symptoms, while Ab tend to appear after 14 days. How were PCR results used ? What PCR reagent was used ?
-
Results of statistical assays are lacking lines 195 and 256. The authors conclude that the two ELISA's are comparable, without providing sufficient evidence. Such conclusion would require larger samples and other tests (Bland-Altman, correlation, % agreement, ...)
-
Figure 3A : please clarify which samples came from infected/vaccinated/healthy patients
-
Figure 3 : It is not possible to conclude on the ability of the two ELISA's to distinguish between patients with a history of COVID-19 and patients without a history of COVID-19 due to overlap between ratios of "infected" and "prior vaccination". Such decrease in anti-spike Ab is not expected after 7 months.
-
Conclusion is only relevant if the objective is clear. Too many aspects, not in line with results, are discussed. For example, what is the link between micro-fluidic assays and POC ? (lab-on-chip described here is not adapted to POC).
-
Conclusion : line 270-271 : evidences not provided. This is beyond the perimeter of this work.
-
Line 286 : advantages versus ELISA for mass testing are not clear (except lower volumes). Shorter hands on time ?
-
Conclusion : potential disavantadges over traditional ELISA are not discussed.
Other minor points should be adressed:
page 5 line 175 : resolution of camera
page 7 line 237 : colorimetric
line 263 : "their use" instead of "its use" would be more correct
line 267 : sentence unclear
lines 258-272 : should be moved to discussion
294 : smartphone is a good point. But microplate readers do more than taking pictures. They associate signals with samples, calculate ratio, manage control and calibration and allow traceability of patients and results. It is doable with a smartphone app but must be taken into account in the context of medical biology.
Author Response
Dear Reviewer,
We appreciate very much your detailed review and all your comments, we understand your viewpoints and we believe that with the requested corrections the manuscript has improved considerably, thank you for your time and experience.
We attach a description of the comments received and our modification in the manuscript:
Point 1: Introduction : it is unclear what it the state of the art regarding SARS-CoV-2 diagnostic using microfluidic devices.
Response 1: In the introduction, we included the most recent microfluidic devices applied in COVID-19 published, briefly describing the sampling capabilities and the volumes reagents.
Point 2: line 86 : The objective of this study is not clearly described. It is unclear if the objective was to develop a lab-on-chip ELISA, to demonstrate its agreement with ELISA results or to demonstrate its advantages over ELISA on plate. It is of limited interest to compare lab-on-chip results to another assay (ELISA) which has not been validated elsewhere.
Response 2: According to your comment, we modified the objective and describe the development of an automated ELISA on-chip.
Point 3: line 122 : 5 healthy patients sampled during COVID-19 pandemic can have anti-spike Ab due to asymptomatic/paucy-symptomatic COVID-19. Therefore, they cannot be used to define average ELISA ratio for healthy patients.
Response 3: To clarify and better understand the samples, we request more information from the hospital that we collaborate, and we attach the kits that were used.
Point 4: page 4 line 124 : PCR cannot be used to confirm serological results because PCR stays positive for 14 days after onset of symptoms, while Ab tend to appear after 14 days. How were PCR results used ? What PCR reagent was used ?
Response 4: To clarify and better understand the samples, we request more information from the hospital that we collaborate, and we attach the kits that were used.
Point 5: Results of statistical assays are lacking lines 195 and 256. The authors conclude that the two ELISA's are comparable, without providing sufficient evidence. Such conclusion would require larger samples and other tests (Bland-Altman, correlation, % agreement, ...)
Response 5: To resolve this point and according to your advice, we generated two Bland-Altman plots to improve the comparison between the immunoassays. Our number of samples is small, but in this communication, we want to present this first stage of the platform, the future is to work with the optimization and considers larger samples.
Point 6: Figure 3A : please clarify which samples came from infected/vaccinated/healthy patients
Response 6: We modify the Figure 3A and 4A and included a code to facilitate the sample interpretation.
Point 7: Figure 3 : It is not possible to conclude on the ability of the two ELISA's to distinguish between patients with a history of COVID-19 and patients without a history of COVID-19 due to overlap between ratios of "infected" and "prior vaccination". Such decrease in anti-spike Ab is not expected after 7 months.
Response 7: To clarify and better understand the samples, we request more information from the hospital that we collaborate, and we attach the kits that were used.
Point 8: Conclusion is only relevant if the objective is clear. Too many aspects, not in line with results, are discussed. For example, what is the link between micro-fluidic assays and POC ? (lab-on-chip described here is not adapted to POC).
Response 8: To clarify, we modify the conclusion section.
Point 9: Conclusion : line 270-271 : evidences not provided. This is beyond the perimeter of this work.
Response 9: According to your comment, we modify the paragraph.
Point 10: Line 286 : advantages versus ELISA for mass testing are not clear (except lower volumes). Shorter hands on time ?
Response 10: According to your comment, we include in the conclusion section information about the disadvantages and the potential of microfluidics and automated assays.
Point 11: Conclusion : potential disadvantages over traditional ELISA are not discussed.
Response 11: According to your comment, we include in the conclusion section information about the disadvantages and the potential of microfluidics and automated assays.
Reviewer 2 Report
The authors in this manuscript presented a novel format for the detection of anti-SARS-CoV-2 antibodies by using an automated immunoassay on-chip, while comparing it to the gold standard methodology utilized for immunoassays. They utilized an automated ELISA on-chip and an image processing software that analyzed a picture taken with a smartphone, a useful alternative when lacking specialized equipment or a laboratory setting.
The manuscript is well written, Results and materials and methods are clearly presented with a good statistical analysis.
The article needs some minor revision:
The number of samples is small, and this is a limitation of the study that should be included in the discussion.
Author Response
Dear Reviewer,
We appreciate very much your review and all your comments, we believe that with the requested corrections the manuscript has improved considerably, thank you for your time and experience.
In this new manuscript we included information in the introduction, discussion and conclusion sections, also we modify the Figure 3 and 4.
Reviewer 3 Report
The manuscript entitled “Automated ELISA on-chip for the detection of anti-SARS-CoV-2 antibodies” demonstrates an automated ELISA on-chip-based method for the detection of anti-SARS-CoV 2 antibodies. Although the authors have presented their study with supporting details, few sections need to be improved prior to recommendation. Here are my comments:
- There were no positive controls mentioned in the manuscript. The authors have mentioned using negative control in figure legends but not in figures. It's highly suggested to add positive controls to improve the confidence in results.
- What are the pros and cons of this automated method compared with the conventional ELISA method? Also please include the limitations of your study in the discussion.
- Please improve the figure legends for ease of understanding. Were the X-axis pertaining to patient sample codes in Figure 2B and 2C? Please clarify. Please also check for all figure legend units.
- Lines 188-196: In processing samples using the automated ELISA on-chip, how many samples can it process for every run and add into text its performance in comparison with conventional ELISA?
- Lines 194-196: The authors have mentioned statistical significances but without adding p-values on the texts. Please add.
- Lines 270-273: The authors have mentioned this method’s future applications, please expound on how it can be utilized further?
Author Response
Dear Reviewer,
We appreciate very much your review and all your comments, we believe that with the requested corrections the manuscript has improved considerably, thank you for your time and experience.
We attach a description of the comments received and our modification in the manuscript:
Point 1: There were no positive controls mentioned in the manuscript. The authors have mentioned using negative control in figure legends but not in figures. It's highly suggested to add positive controls to improve the confidence in results.
Response 1: According to your comment, we include the information and describe the development of an automated ELISA on-chip.
Point 2: What are the pros and cons of this automated method compared with the conventional ELISA method? Also please include the limitations of your study in the discussion.
Response 2: According to your comment, we include information about the disadvantages and the potential of microfluidics and automated assays.
Point 3: Please improve the figure legends for ease of understanding. Were the X-axis pertaining to patient sample codes in Figure 2B and 2C? Please clarify. Please also check for all figure legend units.
Response 3: We think they refer to figure 3B and 3C. We modify the Figure 3A, 3B, 3C and 4A, taking your comment as a reference, we included a code to facilitate the sample interpretation. We generate two Bland-Altman plots to improve our comparative between immunoassays (Figure 3D and 4D).
Point 4: In processing samples using the automated ELISA on-chip, how many samples can it process for every run and add into text its performance in comparison with conventional ELISA?
Response 4: We include the requested information describing
Point 5: The authors have mentioned statistical significances but without adding p-values on the texts. Please add.
Response 5: According to your advice, we include the missing statistical information and Bland-Altman plots information.
Point 6: The authors have mentioned this method’s future applications, please expound on how it can be utilized further?
Response 6: We detailed our future applications and our next works to improve our microfluidic platform.
Round 2
Reviewer 1 Report
I would like to thank the author's for taking some of my remarks into consideration.
In my view, the authors still make unjustified claims regarding their results and Fig 4 is mostly redundant with figure 3. In my view, minor corrections are still required.
Remarks:
- Could the authors please specify in Fig 3D if differences between measurements are "on chip – traditional" or "traditional – on chip" ?
-Fig 3A : Samples 3 and 7 cannot be classified as "vaccinated" as they look like 'pre-vaccination" samples
-Line 121 : "The serum samples provided by the Alfa Medical Center were previously classified using RT-PCR and IgG/IgM tests. ". How authors defined "COVID-19 infected patients" using these assays: positive RT-PCR ? And/or Positive IgG and/or IgM ?
-Line 121: same question as above for "healthy patients"
-English : Line 283 ; because of this "," (please add a coma) : line 311: "their" sensitivity, ...
-Line 243 : the sentence is unclear and can be understood as "there are 23 negative controls and 24 positive controls".
- line 287 : I disagree. No data are provided to support the claim that the test has higher throughput than traditional ELISA, lower cost and easier automation (most ELISA steps can be automated). It is possible but not demonstrated in this study.
- Fig4C and 4D : redondant with Fig 3A an 3D !!
- Line 314 : There is no data supporting the fact traditional ELISA require a more specialized staff than "on-chip" ELISA. Microfluidic technologies do not seem easier to manage than simple washing machines for ELISA plates. Most steps can also be automated with a traditional ELISA.
Author Response
Dear Reviewer,
Thank you so much for your attention and support in improving this manuscript. We are sorry for not fully addressing your comments.
Here is the detail of the corrections made; thank you for your time and experience.
Could the authors please specify in Fig 3D if differences between measurements are "on chip – traditional" or "traditional – on chip" ?
R- We modified the figures 3D and 4D with their respective caption.
Fig 3A : Samples 3 and 7 cannot be classified as "vaccinated" as they look like 'pre-vaccination" samples
R- We totally agree; we modify it.
Line 121 : "The serum samples provided by the Alfa Medical Center were previously classified using RT-PCR and IgG/IgM tests. ". How authors defined "COVID-19 infected patients" using these assays: positive RT-PCR ? And/or Positive IgG and/or IgM ?
-Line 121: same question as above for "healthy patients"
R- We include the detail of the classification process carried out.
-English : Line 283 ; because of this "," (please add a coma) : line 311: "their" sensitivity, ...
R-Thank you, we modify it.
-Line 243 : the sentence is unclear and can be understood as "there are 23 negative controls and 24 positive controls".
R-We decided to include NC and PC to describe the controls and eliminates the sentence. Thank you
- line 287 : I disagree. No data are provided to support the claim that the test has higher throughput than traditional ELISA, lower cost and easier automation (most ELISA steps can be automated). It is possible but not demonstrated in this study.
R-We modified this part according to your comment
- Fig4C and 4D : redondant with Fig 3A an 3D !!
R- We considered it necessary to keep both figures. After all, they are two different methods. The first is the most common by 450 nm absorbance; the second is by colorimetry based on pictures from a smartphone.
- Line 314 : There is no data supporting the fact traditional ELISA require a more specialized staff than "on-chip" ELISA. Microfluidic technologies do not seem easier to manage than simple washing machines for ELISA plates. Most steps can also be automated with a traditional ELISA.
R- According to your comment, we decided to eliminate the sentence.
Reviewer 3 Report
- The manuscript improved after the revision however it could have improved more. Some of the comments were not directly addressed as mentioned in the response.
- Instead of labeling controls as samples 23 and 24, it would have been more clear if it's labeled as PC or NC for clarity.
- The authors have not clearly addressed whether the ELISA-on chip has limitations as a POC device. The authors have only mentioned 'some of the diagnostic challenges related to technical limitations in the existing tests' in Lines 82-83, but have not clearly defined these limitations for readers to understand.
- It was mentioned in the last comments for clarification of Figure 2B/2C legends, whether the X axis in Figure 2B and 2C were patient ids? what does P11.15 mean? Was it a combination of results of Patient 11 and 15 is labeled in Figure 2A?
- The authors did not specify how many samples can be processed per run which the reviewer believes is also a way of comparing traditional and their ELISA on the chip.
Author Response
Dear Reviewer,
Thank you so much for your attention and support in improving this manuscript. We are sorry for not fully addressing your comments.
Here is the detail of the corrections made; thank you for your time and experience.
Instead of labeling controls as samples 23 and 24, it would have been more clear if it's labeled as PC or NC for clarity.
R- We totally agree; we modify it, and include in the figures and captions.
The authors have not clearly addressed whether the ELISA-on chip has limitations as a POC device. The authors have only mentioned 'some of the diagnostic challenges related to technical limitations in the existing tests' in Lines 82-83, but have not clearly defined these limitations for readers to understand.
R- According to your comment, we include the information requested
It was mentioned in the last comments for clarification of Figure 2B/2C legends, whether the X axis in Figure 2B and 2C were patient ids? what does P11.15 mean? Was it a combination of results of Patient 11 and 15 is labeled in Figure 2A?
R- According to your comment, we describe in more detailed the code used in the figures.
The authors did not specify how many samples can be processed per run which the reviewer believes is also a way of comparing traditional and their ELISA on the chip.
R- We include the information, in the line 194.